# Effects of Transitional Care after Hospital Discharge in Patients with Chronic Obstructive Pulmonary Disease: An Updated Systematic Review and Meta-Analysis

**DOI:** 10.3390/ijerph20116053

**Published:** 2023-06-05

**Authors:** Yukyung Park, Jiwon Kim, Sukyoung Kim, Dahae Moon, Heuisug Jo

**Affiliations:** 1Department of Preventive Medicine, Kangwon National University Hospital, Chuncheon 24289, Republic of Korea; 2Department of Elderly Clinical Counseling, Seoul Graduate School of Counseling Psychology, Seoul 03136, Republic of Korea; 3Department of Health Policy and Management, Kangwon National University School of Medicine, Chuncheon 24341, Republic of Korea; 4Department of Internal Medicine, Kangwon National University Hospital, Chuncheon 24289, Republic of Korea; 5Division of Public Health, Kangwon National University Hospital, Chuncheon 24289, Republic of Korea

**Keywords:** chronic obstructive pulmonary disease, transitional care, systematic review, meta-analysis, discharge care plan

## Abstract

This study aimed to systematically review the effects of transitional care programs on healthcare use and quality of life in patients with chronic obstructive pulmonary disease (COPD). Several databases were searched for randomized controlled trials conducted over the past five years, and their quality was evaluated using the Cochrane Risk of Bias 2.0 tool. For indicators with available statistical information, a meta-analysis was performed using RevMan 5.4, and a narrative review was performed for the rest of the results. In the meta-analysis, no statistically significant difference was observed between the intervention and control groups in the number of readmissions and emergency room visits due to COPD. The relative risk (RR) of readmission for COPD was lower in the intervention group. Respiratory-related quality of life tended to be better in the intervention group, though not significantly. Physical capacity was improved in the intervention group. Considering the characteristics of the complex intervention, the context and factors of cases where the expected results could be obtained and cases where the expected results could not be obtained were reviewed and discussed. Based on the results of the analysis, implications for the development of better protocols were presented.

## 1. Introduction

Chronic obstructive pulmonary disease (COPD) is characterized by irreversible and progressive airflow obstruction [1]. As of 2019, an estimated 3,197,000 individuals worldwide have succumbed to COPD [2]. Mortality rates in COPD patients have been found to vary over time and across genders. Previous studies indicated higher mortality rates among males than females from 1994 to 2010 [3]. However, recent research conducted by Mei et al. [4] utilizing the mortality database of the World Health Organization (WHO) for COPD patients in the United States and Europe revealed a consistent decrease in mortality rates among male COPD patients. In contrast, mortality rates among female COPD patients either remained stable or increased. Although the mortality rate attributed to COPD may appear relatively low compared to cardiovascular disease or stroke [5], the challenges of continuous self-management and medical treatment for COPD patients are of paramount importance, given the rise in atmospheric particulate matter (fine dust) and respiratory infections [6,7,8].

Patients with respiratory diseases such as pneumonia, asthma, and COPD who have been re-hospitalized often have difficulties using inhalers, taking medications, and performing daily activities, especially when discharged from long-term care facilities; therefore, the transitional care service (TCS) model was proposed as an alternative [9,10,11]. TCS is emerging as a way to help patients manage their disease at home after they are discharged from the hospital. TCS is a patient-centered treatment model designed to improve the quality of life and treatment of patients with chronic diseases, including COPD, and their families. Moreover, TCS is an intervention method to ensure safe and effective intervention and continue patient management in the process of moving between self-employed and nursing hospitals and long-term care facilities through family education and information sharing among medical staff [12]. According to a study that identified the difficulties experienced by patients in the process of moving to long-term care institutions or elderly living facilities after being discharged from the hospital, the unmet needs of elderly patients, especially at the time of discharge, are related to the diagnosis and treatment plan. Training and information for self-management after discharge and information on social welfare services available near the residence were provided [13,14,15].

Several studies that verified the effectiveness of TCS in patients with COPD evaluated COPD-related readmission and mortality [16,17,18,19]. In contrast, others examined emotional aspects, such as quality of life and emotional function [17,20], and further studies aimed to increase knowledge regarding COPD management [21,22]. Among those that set readmission and mortality as outcome indicators, Benzo et al. [16] provided information on health maintenance, medication planning, and exercise to patients with COPD over the phone after discharge. They found that the readmission rate decreased after six months. Similarly, Ko et al. [17] provided TCS regarding drug use, psychological support, and pulmonary rehabilitation for patients discharged from the hospital for acute exacerbation of COPD. As a result, the readmission rate decreased within one year in the intervention group. In addition, Cotton et al. [18] reported that the hospitalization period and mortality within 50 days decreased by providing acute respiratory distress intervention services to 81 patients with early discharge from COPD. However, the opposite result also exists. In a meta-analysis of studies that evaluated the provision of support for self-management for patients with COPD, Majothi et al. [19] reported that these services had no statistically significant effect on the readmission rate or mortality. Regarding the quality of life and emotional aspects, Ko et al. [17] claimed that the frequency and severity of respiratory symptoms and the degree of activity restricted by dyspnea improved after providing TCS to an intervention group of patients with COPD. Similarly, Johnson-Warrington et al. [20] reported that by providing patients with an aerobic and strength training manual of exercises that can be performed at home, emotional states, such as depression and anxiety, were improved. Among the studies that aimed to increase knowledge of COPD management, Hermiz et al. [21] reported that COPD-related knowledge (role of vaccines, prevention of disease exacerbation, etc.) increased due to initiating a community-based intervention program for patients with COPD. Similarly, Abad-Corpa et al. [22] confirmed that participation in a TCS program improved COPD-related knowledge in patients with COPD. Therefore, the overall effectiveness of TCS for patients with COPD is mixed, especially regarding readmission and mortality rates. This is likely because TCS includes various intervention strategies, and the results will vary according to the components of the intervention or the context of the study. Therefore, precise selection criteria must be defined to synthesize appropriate information from studies evaluating the effectiveness of TCS for patients with COPD, and the mechanisms of the effects should be considered.

Combining the preceding studies, various efforts are being made to improve the quality of TCS to reflect the needs of patients, and measures are being taken to prevent readmission, reduce mortality, and improve the quality of life in the transition process of patients with COPD. Therefore, the importance of TCS is increasing. Due to its complexity and multi-faceted nature, TCS requires high-quality evidence. The existing systematic literature reviews and meta-analyses have two limitations. First, the rate of re-hospitalization, which reflects the need for policies such as the hospital readmission reduction program, was used as the main indicator of the effect of TCS. However, unintended results may arise if the readmission rate is evaluated without considering various factors for patient readmission [23,24]. Second, the most recent systematic review of TCS for patients with COPD only includes data up to the beginning of 2017. As patient-centered policies are increasing and TCS is being provided in various countries and institutions, research related to these topics is also increasing. Therefore, the latest evidence-based research trends, including patient-centered indicators and readmission, must be comprehensively synthesized.

This study aimed to synthesize the effectiveness of TCS on healthcare utilization and health in patients with COPD who were discharged from the hospital. This systematic review includes studies published between January 2017 and December 2021 to analyze the latest available information. Additionally, we tried to discuss meaningful information for improving TCS programs and studies on patients with COPD in the future by broadly synthesizing the factors related to success or failure in the analyzed RCTs. Hopefully, the results of this study will provide useful information for those planning new TCS programs.

## 2. Materials and Methods

This study was conducted according to the Preferred Reporting Items for Systematic Review and Meta-Analysis (PRISMA) guidelines [25], and the formalized protocol was registered in the International Prospective Register of Systematic Reviews (PROSPERO) on 16 March 2022 (registration number: CRD42022307228).

### 2.1. Search Strategy

We searched MedLine, PubMed, Embase, Cochrane Library, CINAHL, and Web of Science from 1 January 2017, to 31 December 2021. Combinations of the following search terms and medical subject heading terms were used: ‘Pulmonary Disease’, ‘Chronic Obstructive’, ‘Pulmonary Disease, Chronic Obstructive’, ‘COPD’, ‘Transitional Care’, ‘Care Transition’, ‘Continuity of Patient Care’, ‘Patient-Centered Care’, ‘Patient Centeredness’, ‘Patient Discharge’, ‘Patient Transfer’, ‘Randomized Controlled Trial’, ‘Controlled Clinical Trial’, ‘Randomized’, ‘Placebo’, ‘Randomly’, ‘Trial’, ‘Groups’. The searched data were managed using Endnote.

### 2.2. Inclusion and Exclusion Criteria

The inclusion criteria for the studies were as follows: (i) RCT or cluster RCT, (ii) adult patients with COPD 18 years of age or older, (iii) interventions consistent with the Transitional Care Model [26,27], (iv) the intervention period starts during patient hospitalization and continues until after discharge, and a face-to-face element is included, (v) the intervention strategy is a multidisciplinary approach with one or more diverse providers, and (vi) studies presented outcome indicators related to health or healthcare utilization.

The exclusion criteria included: (i) non-RCT studies, (ii) several diseases, including COPD, are evaluated in the study with no distinction between the quantitative values of the intervention effect among the diseases, (iii) the intervention consisted of only in-hospital discharge management or community management after discharge in a timely manner, only remote interaction without face-to-face elements, or only a single nurse provider, (iv) the control group did not receive only conventional treatment, (v) the outcome indicators were not health- or medical-related, and (vi) non-English language studies.

### 2.3. Study Selection

From the articles pooled according to the search strategy, we first removed duplicates. Then, two reviewers independently reviewed the literature in the second round. In the first round of screening, we excluded articles that did not meet the inclusion criteria in the title and abstract or met the exclusion criteria. Articles with multiple criteria violations were counted in the following order: participants, interventions, study design, and other criteria. The second screening round reviewed the full text to filter out articles that did not meet the same criteria. In case of discrepancies, by cross-checking the results independently performed at each step, a consensus was reached through discussion. If necessary, a third researcher reviewed the contents to make a final decision.

### 2.4. Data Extraction and Quality Appraisal

Data extraction was performed independently by two reviewers according to the data extraction framework established in advance, and the results were reviewed and synthesized. The contents of the data extraction were as follows: author, year, country, research design, study subjects, monitoring, mediators, intervention factors, outcome indicators, intervention effects, and other key information for evaluating the risk of bias.

The Cochrane Collaboration Risk of Bias 2.0 tool was used to evaluate the risk of bias. The tool consists of five domains, and each domain is subdivided into three to seven questions. The responses to the questions are synthesized into a final value by the algorithm, and the results are presented as low risk, some concerns, and high risk. The risk of bias due to deviations from the intended interventions (effect of assignment to intervention) was analyzed based on the intention-to-treat effect, and a meta-analysis was performed for each outcome indicator. The evaluation was conducted independently by two researchers, and a consensus was reached.

### 2.5. Data Analysis

Among the nine papers included in the systematic literature review, a meta-analysis was performed using the papers that provided the statistical values (mean-standard deviation, relative risk [RR]-standard error) for the indicators using RevMan 5.4 software (the Nordic Cochrane Center, the Cochrane Collaboration, Copenhagen, Denmark). In the analysis of continuous variables, if a pre-post difference value was provided, it was used first; if this value was absent, the post hoc analysis result was used. The indicators eligible for meta-analysis included: (1) the mean difference in the number of COPD readmissions and relative risk of COPD readmission, (2) the mean difference in the number of emergency room (ER) visits, (3) the mean difference in SGRQ (St. George’s Respiratory Questionnaire, respiratory-related quality of life) total score, and (4) standard mean difference of physical capacity (six-minute walk test and the number of steps per day).

To evaluate the heterogeneity, Higgin’s I^2^ statistic and Q statistic were calculated. If the heterogeneity was found to be high, the accuracy of the input data was checked, and the clinical diversity, methodological diversity, and bias were evaluated. If possible, additional sensitivity analysis was performed to identify the cause of heterogeneity. A random effects model was applied to account for the non-homogeneous tendency among studies due to the multi-faceted nature of TCS programs [27,28,29]. Mean difference (MD) was used for indicators that featured the same measurement tool, standardized mean difference (SMD) was used for indicators with different measurement tools, and a 95% confidence interval (CI) was presented for each effect size. The results of the analysis were presented as a forest plot, and the cause of high heterogeneity was identified through sensitivity analysis. We present a narrative review of indicators that met the inclusion criteria but were not used in the meta-analysis.

## 3. Results

### 3.1. Study Selection

A total of 3961 articles were collected from the following databases in the literature search: CINAHL (124 articles), Cochrane Library (608 articles), Embase (1916 articles), Medline (253 articles), PubMed (339 articles), and Web of Science (721 articles). After excluding 980 duplicate documents, 2981 papers were used for screening. According to our exclusion criteria, 2924 articles were not considered in the primary screening (1458 did not meet participants’ criteria, 1247 did not meet intervention criteria, 174 did not meet RCT study design, and 45 did not meet other criteria). In the secondary screening, 48 articles were excluded (one did not meet the participants’ criteria, 14 did not meet intervention criteria, 31 did not meet RCT study design, and two did not meet other criteria). In the end, nine papers were included in the systematic literature review (Figure 1).

### 3.2. Study Characteristics

Table 1 shows the characteristics of the nine papers included in the study. In total, 2002 participants were included in this study, and the number of participants in each study ranged from 42 to 470. TCS was provided in the intervention group. A wide variety of intervention strategies was applied, including customized or standard education (regarding COPD, respiratory exercise, self-management, and drug use and management), discharge plan establishment, personalized action plans, and phone calls or visits after discharge. TCS included enrolling patients during hospitalization, conducting initial evaluations, and providing education or self-management training. After discharge, the intervention was enacted to ensure that education and training provided through periodic follow-ups could be continued. The care plans were written so that the patient could be motivated based on the patient’s preferences and symptoms after discharge. In many cases, the provision of professional education was the basis of TCS. However, many interventions require educator counseling skills, such as communication with patients, psychological intervention, and motivation for self-management. Interventions that reinforce self-management were especially important for patients to continue self-management behaviors after discharge. As an intervention after discharge, some studies provided visit monitoring or made counseling over the phone possible, and periodic visits to the center for follow-up were also utilized.

The intervention period included hospitalization for up to two to twelve months. Although the core intervention provider was generally a nurse, registered respiratory therapists, research assistants, case managers, physical therapists, and respiratory specialists also participated. Readmission was the most common outcome indicator, followed by ER visits, quality of life, respiratory symptoms, and physical capacity. Other indicators not included in the meta-analysis encompassed those related to daily life, such as respiratory symptoms (COPD Assessment Test, CAT), depression and anxiety (Hospital and Anxiety Depression Scale, HADS), sleep, and nutrition. Studies also measured the persistence of the intervention or management behavior.

### 3.3. Risk of Bias

The final result of the risk of bias assessment for each outcome indicator is shown in Figure 2. In the randomization process domain, one paper (11.1%) was judged with some concerns, and in the measurement of the outcome domain, six papers (66.7%) were determined as having some concerns. This item involves the measurement method according to the outcome, the blinding of the outcome evaluator, and whether the outcome affected the knowledge about the intervention. Seven papers used COPD readmission as the outcome indicator; among them, three papers (43%) had a low risk of bias, and four (57%) had some concerns. Four papers used the number of ER visits to judge the outcome, among which three (75%) had a low risk of bias, and one (25%) had some concerns. Furthermore, five papers used the (respiratory-related) quality of life as the outcome indicator, two of which (40%) had a low risk of bias, and three (60%) had some concerns. Lastly, three papers used physical (walking) capacity for evaluation, and all were judged as low risk.

### 3.4. Meta-Analysis Results

#### 3.4.1. Effect on Healthcare Utilization: Readmission and Number of ER Visits for COPD

The average effect of TCS on readmission and ER visits for COPD was meta-analyzed using four papers (Figure 3 and Figure 4). One study analyzed both the number of readmissions and the RR of readmission between follow-up periods. As a result, no statistically significant difference was observed in the number of readmissions between the intervention group and the control group (Mean Difference, MD = −0.15, 95% CI: −1.05–0.74, *p* = 0.74), but the RR of readmission was significantly lower in the intervention group (RR = 0.68, 95% CI: 0.56–0.84, *p* = 0.0004). The number of ER visits did not differ between the two groups (MD = −0.67, 95% CI: −1.95–0.60, *p* = 0.30). Finally, the number of readmissions and ER visits showed significant heterogeneity (Higgin’s I^2^ = 98%), and the RR of readmission was low (Higgin’s I^2^ = 0%).

#### 3.4.2. Effect on Health and Quality of Life: SGRQ Total Score, Physical Capacity

A meta-analysis was conducted using four papers to evaluate the effect on quality of life and three to evaluate the effect on physical capacity (Figure 5 and Figure 6). Aboumatar et al. [30] and Ko et al. [17] provided pre-post mean difference and standard deviation values for the intervention and control groups, respectively, which we used, while Bikmoradi et al. [36], Wang et al. [31], and Granados-Santiago et al. [35] provided only post hoc mean and standard deviation values. The result of the meta-analysis also suggests the respiratory-related quality of life showed, although it was not statistically significant among those four studies (MD = −10.58, 95% CI: −26.48–5.33, *p* = 0.19). TCS was found to have a positive effect on physical capacity, as shown by the improvement in walking ability, although this had weak statistical significance (SMD = 0.56, 95% CI: −0.08–1.20, *p* = 0.08). Considerable heterogeneity was observed in the quality of life (Higgin’s I^2^ = 99%) and physical capacity (Higgin’s I^2^ = 87%).

### 3.5. Sensitivity Analysis

Heterogeneity in the number of COPD readmissions, number of ER visits, quality of life, and physical capacity was high. The complex intervention in TCS may have contributed to the large overall heterogeneity, and the difference among the studies likely led to the large heterogeneity due to the small number of papers used for the meta-analysis.

As a result of checking the papers that caused heterogeneity through sensitivity analysis, when the studies by Aboumatar et al. [30] and Bikmoradi et al. [36] were removed, Higgin’s I^2^ for the quality of life decreased to 34%. Similarly, when the study by Wang et al. [31] was removed, Higgin’s I2 for physical capacity decreased to 0%. However, the heterogeneity of the number of COPD-related readmissions and ER visits did not change when any studies were removed from the analysis. The study by Aboumatar et al. [30], which showed the most heterogeneity in the analysis of various outcome indicators, cited the characteristics of the study subjects as a limitation. Specifically, patients with comorbidities or poor health behaviors, such as smoking or those requiring continued oxygen therapy, were excluded; while a high number of patients with a low socioeconomic status, which might have contributed to the negative results, was included. However, the number of readmissions and ER visits, which remained highly heterogeneous even when this paper was excluded, shows the possibility of raising the issue of the feasibility of using healthcare utilization as a primary indicator of TCS effectiveness.

### 3.6. Systematic Review of Other Outcomes and Related Factors

The effect of TCS on readmission due to COPD exacerbation was evaluated in seven of the nine studies. Due to the limited verifiable information, the results of only three studies could be pooled for the meta-analysis of the number of readmissions, and the results of two studies were used to find the RR of readmission. The analysis results showed that TCS had different effects on different readmission-related indicators, such as the mean number of readmissions and the RR of readmission. Regarding readmission, meta-analyses that use a single indicator are limited in interpretation because various indicators are present, including the number of readmissions, readmission rate, time to first readmission, intensive care unit use, or length of hospitalization during readmission. Five [17,31,32,34,37] of the seven studies showed a lower risk of readmission in the intervention group, while one [33] study showed no statistically significant difference. In the remaining study [30], the intervention group had a higher risk of readmission than the control group.

The effect on ER visits was calculated in four studies. In one study, the number of ER visits due to COPD was significantly lower in the intervention group [31], while in two studies [34,37], no statistically significant difference was observed. In the remaining study, the number of ER visits was higher in the intervention group [30].

Respiratory-related quality of life (SGRQ) was used as an evaluation indicator in five studies. Three studies [17,31,36] reported that the intervention group had statistically improved respiratory-related quality of life, and the remaining two studies reported no statistically significant difference [30,37]. Two studies also described the overall health-related quality of life: one showed significant improvement in the intervention group [35,38], while the other showed no significant difference between the two groups [37].

Although COPD symptom assessment (CAT) was used in two studies, a meta-analysis could not be performed due to the lack of information; however, both studies showed statistically significant improvement in the intervention group [32,33]. Similarly, a study by Ko [17] included lung function and modified Medical Research Council (mMRC) dyspnea score indicators as objective and subjective respiratory symptom indicators, respectively. They found no significant difference in lung function, but mMRC was significantly improved in the intervention group.

Physical capacity (six-minute walk test, number of steps per day) was evaluated in three studies. A significant improvement was observed in the intervention group in two studies [31,35], and no statistically significant difference was observed in one study [17].

Depression and anxiety (HADS) were included in two studies, but a meta-analysis could not be performed due to the lack of information. The intervention group did not show a statistically significant improvement compared to the control group in either study [32,37].

Although many studies have focused on healthcare utilization, respiratory-related quality of life, and symptoms as indicators of the effectiveness of TCS interventions in patients with COPD, some studies emphasized patient-centeredness, focusing on the patient’s management capabilities and daily life. Granados-Santiago et al. [35] included patients’ knowledge of COPD, inhaler compliance, and general function in their evaluation, all showing statistically significant improvement in the intervention group. The study by Rose et al. [37] included self-efficacy, patient satisfaction, caregiver burden, smoking cessation, and vaccination status as indicators for COPD management, all showing no statistically significant difference between the two groups.

Of the studies that met the inclusion criteria, factors that contributed to the success or failure of transitional care was narratively synthesized. TCS may not have affected readmission as well as expected in the intervention group for several reasons. First, according to the baseline survey, many participants in the intervention group were men, who may have had a greater tendency to visit the ER than women due to a lack of experience in seeking ambulatory care and using routine care services [30,39,40]. Second, patient activation was improved due to the motivation improvement intervention; therefore, they were better suited to communicate with medical personnel, resulting in earlier recognition of worsening symptoms and active healthcare use. Third, the intervention group had a high prevalence of substance abuse problems, suggesting that the management recommended by the intervention may not have been followed [30,33]. Furthermore, while pulmonary rehabilitation significantly reduces readmission, the degree of improvement may have been limited because the hospital was not available to provide pulmonary rehabilitation during the study [17,33,41]. Regarding the study design, the monitoring period during which significant changes can be expected may vary depending on the characteristics of the indicators. For example, indicators such as quality of life require long-term monitoring to detect a change due to intervention; therefore, the results may not be adequately evaluated by short-term monitoring. Conversely, indicators related to behavioral change, such as disease management, are significant in short-term monitoring, but the effect disappears in long-term monitoring [31,42]. In addition, several studies have reported significant dropout rates during monitoring, suggesting that various approaches are needed for prevention [33]. Similarly, Hegelund et al. [32] reported that the process of establishing a patient-led, individualized discharge plan comes as an excessive burden to patients and leads to dropout.

However, TCS was effective for the following reasons. Firstly, support through constant phone calls, visits, and technical training and education may have helped maintain patients’ physical activity [31]. Secondly, unlike other interventions that only include the provision of booklet educational materials or discharge plans, interventions aimed at exercise education and practice were more successful in improving quality of life indicators through improving physical capacity in patients with COPD [31]. Thirdly, interventions that provide personalized action plans after discharge help patients objectively recognize the seriousness of their condition through periodic CAT evaluations [32]. Finally, intensive education and personalized case management interventions that support patients to play an active role in changing health behavior based on the patient’s self-determination competency were cited as important factors that can significantly impact the outcome [34,35].

Regarding the discharge transition management program, the coordinator’s competency is important because the active management of the patient and the interaction with the coordinator play a vital role. Therefore, for successful intervention, a standard education program, continuous refresher education, and quality management of intervention are needed to ensure that the variation in competency among coordinators is not severe. Additionally, Hegelund et al. [32] suggested that care must be taken to avoid placing excessive responsibilities or burdens on patients.

## 4. Discussion

This paper analyzed the average effects of TCS on health and healthcare utilization in patients with COPD based on the transitional care model by synthesizing RCT studies with the highest level of evidence. This study analyzed the results of more recent research than that used in the previous systematic literature reviews. Additionally, as the outcome indicator, we aggregated the effects of various indicators related to patient health and healthcare use.

According to the results of the meta-analysis, among the available types of indicators, in terms of healthcare utilization variables, there was no significant difference in the average number of readmissions and emergency department visits, with only the relative risk of readmission showing a statistically significant reduction. In terms of health and quality of life indicators, respiratory-related quality of life showed a positive trend, although not statistically significant, and walking ability (physical capacity) showed a statistically significant improvement, suggesting a relatively modest effect compared to healthcare utilization indicators. A study by Ridwan et al. [43] and Liu et al. [44], which analyzed the effectiveness of transitional care in patients with COPD, validated a statistically significant reduction by analyzing the odds ratio and relative risk of readmissions for COPD. This is consistent with our relative risk results, but they did not present an analysis of the number of readmissions, so we were unable to make a comparison. Readmission and ER visits were frequently used as performance indicators for TCS-related policies. Readmission is often used as a key performance indicator, as it indicates the quality of medical care and the degree of financial burden due to unnecessary healthcare utilization. However, according to the results of a systematic literature review of recent studies, caution is advised for researchers considering simply synthesizing the effects of readmission and ER visit indicators.

For several reasons, determining the effect of TCS on healthcare indicators was not a simple process. Firstly, the diversity of indicators was challenging. Outcome indicators for readmission and ER visits were expressed by various parameters, such as the rate of visiting more than once, the rate of never visiting, the time to readmission, and the number of visits/readmissions; each indicator related to readmission had a different effect. For example, in the study by Silver et al. [34], the number of ER visits or readmissions due to COPD was not statistically significant. However, the probability of multiple readmissions or ER visits was lower, the probability of being admitted to the ICU when readmitted was lower, and the length of stay was shorter in the intervention group. The authors explained that this might be because the intervention group was quickly hospitalized and treated or managed well at home before their condition worsened due to the effectiveness of the intervention.

Re-hospitalization is caused by various mechanisms in addition to post-discharge management. Depending on the characteristics of the country’s health and welfare system, patients may not have anyone to take care of them at home after discharge, so they may be hospitalized (social admission). In other cases, according to the payment compensation system, patients are induced to be discharged quickly before they achieve medically adequate recovery.

Unlike the healthcare utilization indicators, the health-related indicators such as disease-related symptoms, respiratory quality of life, and physical capacity tended to improve with TCS. This suggests that TCS for patients with COPD may be a meaningful intervention in terms of enhancing the patient’s disease management skills, even if its ability to reduce additional healthcare utilization may be limited. Even when trying to establish TCS for discharged patients as a policy, care must be taken to set evaluation indicators only for healthcare use, such as readmission, and patient-centered indicators must be prioritized.

Our study is strengthened by its inclusion of the most recent studies of transitional care interventions for patients with COPD. In addition to readmission, which has been the focus of attention, the effects of various indicators emphasizing patient-centeredness were widely included, and the results were synthesized. Nonetheless, this study had some limitations. First, compared to other systematic reviews, fewer studies among those screened met our inclusion criteria for this review. Since transitional care comprises bundles of various interventions, many heterogeneous studies are inevitably retrieved during the literature searches on the subject. Due to our strict inclusion criteria imposed in order to conduct a high-quality meta-analysis, fewer studies were deemed appropriate for synthesis in this case. Second, while heterogeneity was high due to the complex nature of intervention in TCS, additional sub-analysis could not be performed because the number of studies available for meta-analysis was not large for each indicator. Instead, we evaluated various indicators to sufficiently describe the factors of success and failure of interventions that resulted in different outcomes depending on the interventions and context of each study.

## 5. Conclusions

This systematic review suggests that multi-component TCS for patients with COPD may affect healthcare outcomes. However, more attention should be paid to the selection of indicators and implementation of interventions to achieve targeted outcomes. The results of the meta-analysis also suggest that TCS can help improve patients’ respiratory-related quality of life and physical capacities. Based on the success and failure factors reviewed in the studies included in the systematic review, the implications for developing better TCS protocols going forward are as follows.

Firstly, the components of intervention included in TCS varied across studies but commonly included customized discharge planning, disease management education, and monitoring based on the patient’s individual status. Respiratory rehabilitation is often included in usual care and is known to be effective in managing patients with COPD; however, accessing and sustaining respiratory rehabilitation is challenging [45,46,47]. Therefore, when providing TCS, it would be beneficial to include a goal of reducing barriers that prevent patients from starting or maintaining respiratory rehabilitation for various reasons. Follow-up management differed in quantity and quality depending on the study. According to the success/obstacle factors, follow-up management significantly affected the continuation and maintenance of health-related behaviors. It is suggested that qualified follow-up management should include more than simply evaluating monitoring indicators.

Secondly, indicators should be selected in detail in consideration of intervention factors. Even if the goal is to reduce readmissions, based on the results of existing studies, specific indicators should be set considering the mechanism by which the intervention works. In addition, it should be considered an intervention period and a monitoring period when selecting indicators.

Finally, a system that can sufficiently enhance the competence of the coordinator must be established. In particular, to provide patient-centered transition management interventions, knowledge of medical and nursing management of COPD, as well as patient-tailored education skills and counseling approaches that can elicit the patient’s intrinsic motivation, are needed. Significant efforts should be made in organizing, training, and quality control of the training program for interventionists before RCTs are performed.

## Figures and Tables

**Figure 1 ijerph-20-06053-f001:**
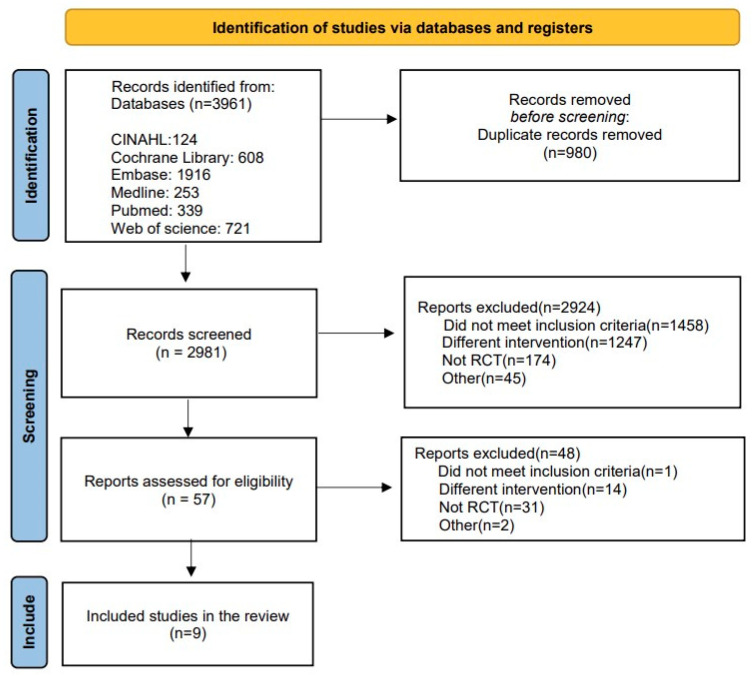
PRISMA flowchart of the study selection procedure.

**Figure 2 ijerph-20-06053-f002:**
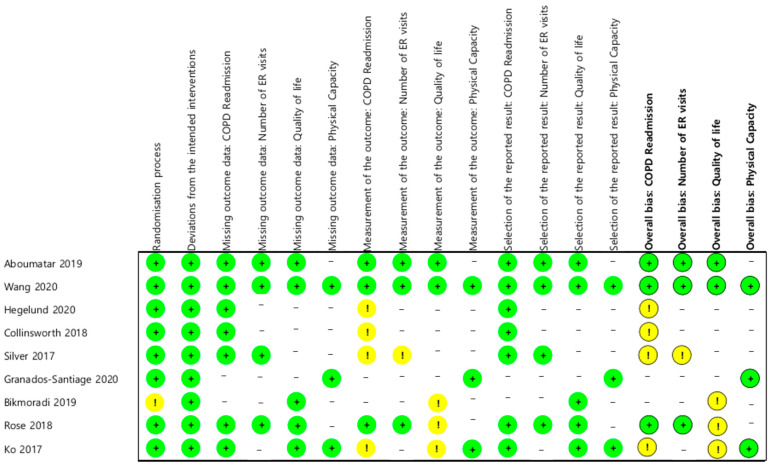
Risk of bias for outcomes [17,30,31,32,33,34,35,36,37].

**Figure 3 ijerph-20-06053-f003:**
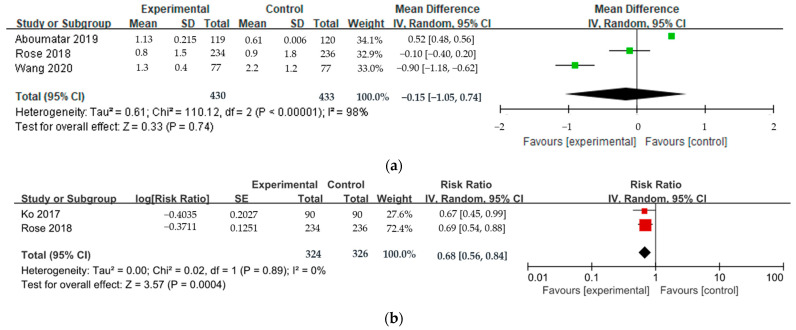
Effect of transitional care on readmission for COPD. (**a**) Number of readmissions; (**b**) Relative risk (RR) of readmission.

**Figure 4 ijerph-20-06053-f004:**
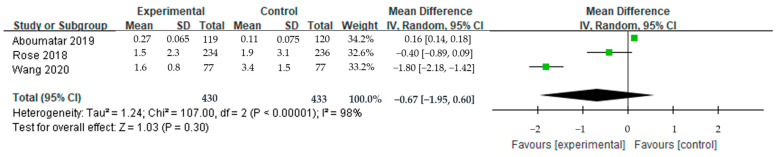
Effect of transitional care on the number of emergency room (ER) visits for COPD.

**Figure 5 ijerph-20-06053-f005:**
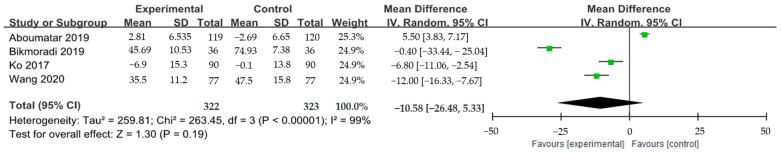
Effect of transitional care on quality of life (St. George’s Respiratory Questionnaire).

**Figure 6 ijerph-20-06053-f006:**
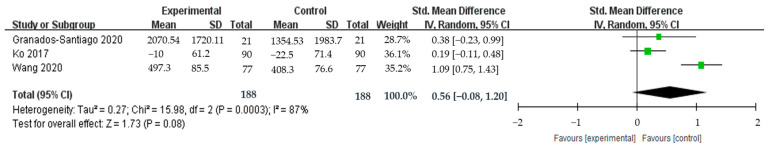
Effect of transitional care on physical capacity.

**Table 1 ijerph-20-06053-t001:** Study characteristics.

Author (Year) Country	Study Design	Participants	Intervention Group	Control Group	Components of the Intervention	Duration of Intervention	Coordinator	Main Outcome
Aboumatar et al. (2019) USA [30]	RCT (blinded data collectors and outcome assessors)	240 IG: 120 CG: 120	Combined transitional care and long-term self-management Support	Usual transitional care	(i) Customized transition support services (ii) Self-management training and support (iii) Community programs and treatment service support (Patient-centered partnership approach)	From hospitalization to 6 months after discharge	COPD nurse	Readmission, emergency department visits, quality of life, death
Wang et al. (2020) China [31]	RCT (blinded interventionists)	154 IG: 77 CG: 77	Nurse-led self-management program	Usual care, health education for COPD	(i) Comprehensive patient assessment (ii) Five or six face-to-face education sessions before discharge (individually tailored) (iii) Discharge planning for each participant (iv) Three-month follow-up intervention	From approximately 6–7 days before discharge to 3 months after discharge	Advanced respiratory nurse	Readmission, emergency department visits, quality of life, physical capacity
Hegelund et al. (2020) Denmark [32]	RCT (not blinded)	99 IG: 49 CG: 50	Personalized action plan	Usual care and treatment	(i) Personalized action plan (check status according to the CAT value measured periodically and suggest actions to be taken) (ii) Self-management dialogue, including short instruction and the possibility for subsequent support	From hospitalization to 3 months after discharge	Respiratory-educated study nurses	Readmission, symptoms assessed, anxiety and depression
Collinsworth et al. (2018) USA [33]	RCT (waiver of patient consent for blinding)	308 IG: 141 CG: 167	Patient education and shared decision-making	-	(i) Baseline assessment (ii) Pragmatic COPD Chronic Care education program (iii) Shared decision-making-based self-management planning (iv) Telephone follow-up (1 wk, 1, 2, 6 mo) using a structured checklist	From hospitalization to 6 months after discharge	Registered respiratory therapist	Readmission, symptoms assessed, patient activation measure
Silver et al. (2017) USA [34]	RCT (blocked randomization)	428 IG: 214 CG: 214	Disease management program	-	(i) Education based on Global initiative for chronic obstructive lung diseases(GOLD) guidelines (ii) Individualized written action plan (iii) Scheduled telephone monitoring (Q&A and consultation)	From hospitalization to 6 months after discharge	Respiratory therapist	Readmission, emergency department visits, all-cause mortality
Granados-Santia go et al. (2020) Spain [35]	RCT	42 IG: 21 CG: 21	Shared decision-making and patient engagement program	Standard treatment	Personalized shared decision-making patient engagement program	From hospitalization to 3 months after discharge	-	Physical capacity, health status, knowledge of the disease, pharmacological management, general functionality, nutritional status
Bikmoradi et al. (2019) Iran [36]	RCT	80 IG: 40 CG: 40	Continuous care	Usual care	(i) Four intervention sessions (awareness, sensitization, control, and evaluation) with education package (ii) Telephone follow-up (iii) Invite to LDBE Hospital education orientation for continuous care	From hospitalization to 2 months after discharge	Researchers’ assistants, Hospital Respiratory Center	Quality of life
Rose et al. (2018) Canada [37]	RCT (blinded inspection research assistant)	470 IG: 234 CG: 236	Program of integrated care	Usual care	(i) Standardized education session (ii) Individualized care and action plans (iii) Telephone consultations, action plan, teach-back sessions (iv) Ongoing case manager communication with family physicians and hospital specialists, including respirologists (v) Priority access to ambulatory outpatient clinics and exacerbation management prescriptions	From hospitalization to 3 months after discharge	Case manager	Readmission, emergency department visits, mortality rate, quality of life, anxiety and depression, self-efficacy
Ko et al. (2017) Hong Kong [17]	RCT (blinded examination and investigation research assistants)	180 IG: 90 CG: 90	Comprehensive care program	Usual care	(i) Two educational sessions by a respiratory nurse (ii) The physiotherapist provided every patient with an individualized physical training program (iii) Respiratory physician prescribed medications (iv) Provided consultation phone number available during work hours (v) Follow-up at respiration center every three months	From hospitalization to 12 months after discharge	Respiratory nurse, physiotherapist, respiratory physician	Readmission, quality of life, mortality rate, physical capacity, lung function

IG, intervention group; CG, control group.

## Data Availability

The data that support the findings of this study are available from the corresponding author upon reasonable request.

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
