# Peer review of "Effects of Transitional Care after Hospital Discharge in Patients with Chronic Obstructive Pulmonary Disease: An Updated Systematic Review and Meta-Analysis"

_ijerph, 2023, doi:10.3390/ijerph20116053_

Round 1

Reviewer 1 Report (Previous Reviewer 3)

Dear Authors,

Second review of article I appreciate and highlight the effort, on the part of the authors, to improve the article. It is appreciated that they have considered the suggestions made.

Author Response

We wrote an answer to the comment and attached it as a file. We appreciate your valuable time and opinion for the advancement of this paper.

Reviewer 2 Report (New Reviewer)

1. Clearly explain how they made the selection of articles according to PRISMA

2. At the beginning of the article it is stated that the available systemic reviews consider readmission of the patient to hospital as a marker to evaluate TCS, which they do not consider adequate; however the authors consider using readmission. Readmission can be a parameter affected by multiple patient variables as mentioned, why do you consider it?

3. The inclusion criteria should be clearer so that they are consistent with the objective and conclusions of the study

4. The objective of the study is partially fulfilled since they manage to evaluate the effectiveness of TCS but not the quality of life of the patient.

Author Response

We wrote an answer to the comment and attached it as a file. We appreciate your valuable time and opinion for the advancement of this paper.

Reviewer 3 Report (New Reviewer)

This paper describes a meta-analysis of different interventions aimed to support the transition of the patients from hospital to home on their impact on readmissions, ER visits, quality of life and six minutes walking distance.

1.       the main problem with this meta-analysis resides in the great heterogeneity regarding the intervention (type and length) and the characteristics of the participants among the different studies. heterogeneity is so high, that one important concern is about the validity of the results, i.e. The pertinence of pooling together studies that, while addressing the same topic, are so different.

2.       The authors use a random effect analysis, nonetheless they wrongly consider their summarizing estimations as estimates of average effects. Unfortunately, it is not uncommon to interpret meta-analysis results in the same manner regardless of whether a fixed effect or random effects model is used. In random effect meta-analysis when confidence intervals do not contain zero, the only warranted conclusion is that there is strong evidence that, on average, the treatment effect is beneficial.  BMJ 2011;342:d549.

3.       The discussion is unfocused. In general terms the discussion should contain a brief description of the main results related to the objectives of the study, a comparison of the results of the work with similar published information, the strengths and limitations of the work, the relevance of the study for the scientific community or clinical practice, and finally the conclusions. However the authors opt for a lengthy report of their results, with plenty of speculative explanations about why they were as they were, including, comparisons with rehabilitation and so on.

Specific

1.       Line 33: “Chronic obstructive pulmonary disease (COPD) is caused by chronic inflammation 34 of the bronchi and lung tissue due to smoking, indoor and outdoor air pollution, and 35 respiratory infections [1]. COPD now affects millions of people worldwide nowadays, 36 the number of COPD-related deaths has rapidly increased in recent years, and this dis-37 ease ranked third on the WHO list of the top 10 causes of death in 2020 [2]. COPD is an 38 irreversible disease that requires continuous self-management and medical treatment to 39 maintain an optimally functional state under conditions of limited lung function, to 40 maintain independence and to reduce readmission and mortality due to acute exacerbation [3]. When COPD progresses acutely, 50% of hospitalized patients die within 3.3 42 years, and 75% die within 7.7 years [4]. Dyspnea, a typical symptom of COPD, greatly 43 impacts patients’ daily life. To avoid shortness of breath, patients reduce their daily ac-44 tivities, which worsens their respiratory distress due to physical reconditioning, leading 45to a vicious cycle in which”  This section consists of general information unrelated to the topic of the paper. The authors should eliminate this general information from the introduction. The aim of the introduction is to discuss the rationale of the work, namely the information or lack of information justifying they need of the investigation.  Bay the way the mortality of COPD shows a decreasing trend (even if slower than other big killers, what makes its mortality to go up in the ranking) Mei F et al Chronic obstructive pulmonary disease (COPD) mortality trends worldwide: An update to 2019. Respirology. 2022 Nov;27(11):941-950. doi: 10.1111/resp.14328.  2: López-Campos Mortality trends in chronic obstructive pulmonary disease in Europe, 1994-2010: a joinpoint regression analysis. Lancet Respir Med. 2014 Jan;2(1):54-62. doi: 10.1016/S2213-2600(13)70232-7.

2.       This reader finds unconventional to quote the Daily Mirror (ref 2), given that it is not a peer-reviewed publication.

3.       Line 98: “Several studies have conducted systematic literature reviews and meta-analyses on 98 papers that evaluated the effectiveness of TCS for patients with COPD. For example, a 99 systematic literature review was conducted on 170 studies that provided TCS for patients100 with COPD, and 13 papers were finally derived. The meta-analysis found TCS to be ef-101 fective in reducing all-cause and COPD-related readmission [21]. In addition, Liu et al. 102 [22] conducted a systematic literature review of 409 studies on the provision of TCS to 103 reduce readmission among patients with COPD from 1990 to April 2016, and 7 final pa-104 pers were derived. The meta-analysis confirmed that the COPD-related readmission rate 105 was decreased after 6 and 18 months.” The authors shouldn't include what seems to be part of their results in the introduction.

4.       Line 137: this reader wonders why the authors did not include terms such as telemedicine, tele-monitorization, telehealth or mobile health technology, among others related to the same topic, in their review. Quite a few of the studies of telemedicine are aimed to provide transitional and post admission healthcare.

5.       Line 188: “(mean, 188 relative risk [RR], standard deviation) this is confusing. Mean and standard deviation should be together and relative risk plus a standard error should go next.

6.       191: “if a pre-post difference value that”. The authors need to specify how they calculated the effect size i.e. If by using control groups standard deviation, intragroup standard deviation, or any other method. By the way, it would be better if the authors first name the variables and then the statistic summarizing it; for example, mean differences for six minute walking distance or relative risk for ER visits.

7.       Line 200: “A random effects model was applied to 200 evaluate the degree of heterogeneity” Random effect models do not allow to evaluate the degree of heterogeneity, rather is an approach when the studies are heterogeneous.

8.       line 274: “but the RR of readmission was significantly lower in the intervention group (RR = 0.68, 95% CI: 0.56–0.84, p = 275 0.0004). I must remember the authors that when a random effect model is used they assume that the mean estimate is not the effect of the treatment, but the average effect of the different interventions. with an I2 of 98%, this reader even wonders if pulling together search different studies is meaningful

9.       Line 413 “TCS may not have effected readmission as well as expected in the intervention 413 group for several reasons. First, according to the baseline survey, many participants in 414 the intervention group were men, who may have had a greater tendency to visit the ER 415 compared to women due to a lack of experience in seeking ambulatory care and using 416 routing care services” Any published evidence of this?

Author Response

We wrote an answer to the comment and attached it as a file. We appreciate your valuable time and opinion for the advancement of this paper.

This manuscript is a resubmission of an earlier submission. The following is a list of the peer review reports and author responses from that submission.

Round 1

Reviewer 1 Report

The article Systematic Review Effects of transitional care after hospital discharge in patients 2 with Chronic obstructive pulmonary disease: An updated sys-3 tematic review and meta-analysis” highlights the role of Transitional care service (TCS) in COPD care. The authors also suggest that multi-component TCS for patients with COPD may affect healthcare outcomes. I have read the paper with interest and feel that it is relevant for the area of COPD care and prognosis.

I suggest a few minor revisions. Comments are made below regarding the article.

  1. The authors should clarify in the introduction the aim and objectives of the systematic review.
  2. Minor punctuation and spelling require

Reviewer 2 Report

This systematic review addresses a topic of great interest, regarding the usefulness of transitional care after hospital discharge in COPD.

However, while a large number of papers were included in the first screening, only a few were useful for the analyses, which do not seem sufficient to draw significant conclusions. 

Reviewer 3 Report

Dear Authors,

I added manuscript with comments and modification suggested.

By other way,  I fell that outcome presented are lazy... since is very difficult support any conclusion with 2 and 3 articles reviewed. I understand that heterogeneous of each study.

Recommend rewrite the manuscript in agree solid outcomes.
